# Ischemic Stroke Disrupts Sleep Homeostasis in Middle-Aged Mice

**DOI:** 10.3390/cells11182818

**Published:** 2022-09-09

**Authors:** Rishi Sharma, Abigail Chischolm, Meet Parikh, Adnan I. Qureshi, Pradeep Sahota, Mahesh M. Thakkar

**Affiliations:** Harry S. Truman Memorial Veterans Hospital and Department of Neurology, School of Medicine, University of Missouri, Columbia, MO 65201, USA

**Keywords:** insomnia, sleep homeostasis, mice, stroke, middle cerebral artery occlusion, delta, theta, MCAO

## Abstract

Sleep disturbances, including insomnia and excessive daytime sleepiness, are highly prevalent in patients with ischemic stroke (IS), which severely impacts recovery and rehabilitation efforts. However, how IS induces sleep disturbances is unclear. Three experiments were performed on middle-aged C57BL/6J mice, instrumented with sleep recording electrodes and/or subjected to 1 h of middle cerebral artery (MCAO; Stroke group) or sham (Sham group) occlusion to induce IS. After 48 h of reperfusion (a) experiment 1 verified sensorimotor deficit (using Garcia scale) and infarction (using TTC staining) in this mouse model; (b) experiment 2 examined the effects of IS on the quality (sleep latency and NREM delta power) and quantity (duration) of sleep; and (c) experiment 3 determined the effects of IS on sleep homeostasis using sleep deprivation (SD) and recovery sleep (RS) paradigm. Stroke mice display (a) a significant correlation between sensorimotor deficit and cerebral infarction; (b) insomnia-like symptoms (increased sleep latency, reduced NREM duration and delta power) during the light (inactive) period and daytime sleepiness-like symptoms during the dark (active) period mimicking sleep in IS patients; and (c) impairments in the markers of sleep pressure (during SD) and sleep dissipation (during RS). Our results suggest that IS disrupts sleep homeostasis to cause sleep disturbances.

## 1. Introduction

A stroke occurs either due to an interruption in the arterial blood flow supplying blood to some (focal) or all regions (global) of the brain, termed as ischemic stroke (IS), or due to rupturing of blood vessels, also known as hemorrhagic strokes. However, the majority of strokes (85%) are ischemic in nature [1]. Indeed, ischemic stroke is the fifth leading cause of death, a highest contributor to the disability, and a huge economic burden in the U.S [2,3]. It affects >13.5 million people and kills >5 people annually [4,5,6,7]. The majority of ISs occur due to thromboembolic plaques in the middle cerebral artery (MCA), leading to long-term sensorimotor deficit [8]. In addition, people in the middle-age age range (40–60 years old) are specifically vulnerable to stroke occurrence and re-occurrence [9].

Patients suffering from strokes display a multitude of symptoms, including hemiplegia or hemiparesis, dysphagia, and cognitive impairments. However, the most common, persistent, and debilitating symptom observed in stroke patients is sleep disturbances, including insomnia during sleep period (night) and intense daytime sleepiness [10,11]. In addition, sleep disturbances are associated with worse motor outcomes and slower functional recovery [12,13]. In contrast, sleep enhancement is associated with faster recovery and promotes neurorehabilitation following clinical [14,15] and experimental [16,17,18,19] strokes. Thus, sleep disturbances are considered to be a unique, novel, and modifiable treatment target that can potentially enhance recovery and rehabilitation efforts in stroke patients [20,21]. To understand how IS affects sleep, and develop treatment strategies, it is important to understand the underlying sleep regulatory mechanism that is affected by IS.

Sleep is regulated by two distinct but interrelated processes: homeostasis and circadian. The homeostatic process (Process S) regulates quality (delta or slow wave (SWA) activity) and quantity (duration) of sleep, whereas the circadian process (Process C) regulates the timing of sleep [22]. Since sleep homeostasis is a major mechanism that controls both the quality and quantity of sleep, we focused on understanding the effect of IS on sleep homeostasis. We hypothesized that IS disrupts sleep homeostasis to cause sleep disturbances.

To test our hypothesis, we used adult, middle-aged C57BL/6J mice and intraluminal 1 h MCA occlusion (MCAO) to induce IS. We first verified whether the sleep disturbances observed in these animals mimic human sleep disturbances, including insomnia during the sleep period and sleepiness during the active/wake period observed after IS. Next, we evaluated whether sleep homeostasis is disrupted following IS.

## 2. Materials and Methods

### 2.1. Animals

Adult, 10–12 months old (equivalent to middle-aged humans), male C57BL/6J mice were used in this study. Mice were bred locally at the Harry S. Truman Memorial Veterans Hospital vivarium (Columbia, MO, USA; SAS # 233; IACUC # 9606). Animals were transferred to the experimental room from the vivarium and housed (four per cage) under ambient room temperature (24 ± 1 °C), light (12 h)/dark (12 h) cycle (lights ON at 10:00 a.m.), and ad libitum access to food and water. Animals were allowed to habituate these conditions (at least 2 weeks) before any surgical procedure or experiment was performed. We made every possible effort to reduce suffering to the animals and, also, the number of animals used. All experiments used in this study were approved by local committees (Protocol #233) and performed according to the Association for Assessment and Accreditation of Laboratory Animal Care policies and the Guide for the Care and Use of Laboratory Animals.

### 2.2. Experimental Design and Statistics

Online Graphpad randomization calculator was used for group assignments just before the experiment. The “R” statistical software (version 3.6.3, R Core Team, Vienna, Austria) was used to perform all the statistical analysis [23]. The online Graphpad’s Outlier calculator (Grubb’s) was used to identify the possible outlier. The a priori power analysis (N = 3/group; α = 0.05; power ≥ 0.9; G*Power [24]) was performed to calculate the effect and sample size. Kolmogorov–Smirnov test was performed to determine whether the data were normally distributed. Three experiments were performed with the following exclusion criteria: (1) death after MCAO surgery prior to experimental endpoint; (2) deterioration in the general condition and spontaneous behavior; (3) operation time > 20 min; (4) animals with baseline asymmetry in limb usage, (5) signs of subarachnoid hemorrhage during brain sampling; and (6) IS animals with <6 and >12 Garcia score (based on the results of experiment 1).

### 2.3. Experiment 1

Experiment 1 was designed to validate and verify the right MCAO model in middle-aged mice using behavioral (sensorimotor performance) and histological (infarction) assessment. This was to minimize variability in the effects of MCAO on sleep parameters (experiments 1 and 2), which may originate from variable MCAO outcomes [25]. The animals (N = 6) were subjected to MCAO during the middle of the light period, as previously described [26,27]. Briefly, under isoflurane (1.5%; Baxter, Deerfield, IL, USA) anesthesia, a mouse was placed in the supine position with continuous monitoring and maintenance of normal body temperature by a thermostat-controlled heating pad. Subsequently, under aseptic conditions, the right common (CCA), external (ECA), and internal (ICA) carotid arteries were exposed using an operating microscope. The ECA was tied and cut off to obtain access to the free end of the ECA. A 6-0 silk suture was used to tie a temporary knot around the ICA and CCA to prevent the bleeding once an incision was made on the ECA. A 6-0 nylon monofilament, coated with silicon hardener mixture on one end (Doccol Corp., Redlands, CA, USA), was inserted in the ECA after partial arteriotomy and pushed it further into the ICA (after opening the knot) to occlude the origin of the MCA in the circle of Willis. Following MCA occlusion, the silk suture was tightened around the ECA stump to secure the intraluminal nylon filament and prevent bleeding, and then open the knot on the CCA slowly before removing it to check for bleeding. After 60 min, the monofilament was retracted after a temporary knot was tightened around the CCA to prevent bleeding. Once the monofilament was completely removed, the ECA was cut prior to the bifurcation with ICA using a cauterizer (to prevent bleeding). Next, the knot on the CCA was opened to initiate reperfusion. The surgical wound was moistened with several drops of sterile saline and closed the incision layers (dermis, panniculus carnosus, subcutaneous tissue) with 6-0 PDS absorbable monofilament suture using a simple interrupted pattern. Carprofen (5 mg/kg) was administered subcutaneously. After the surgery, animals were allowed to recover from surgical stress and left undisturbed until examined for sensorimotor deficit and brain infarction on day 2, as described in the following.

#### 2.3.1. Assessment of Sensorimotor Deficit

Sensorimotor deficits were determined using the Garcia scale [28] modified by Yamauchi et al. [29] for mice. This scale involves six parameters to assess the sensorimotor deficit including (1) spontaneous activity, (2) symmetry in the movement of the four limbs, (3) forepaw outstretching, (4) climbing ability, (5) body proprioception, and (6) response to vibrissae touch. Each aspect was scored on a scale of 0 (maximal deficit) to 3 (normal). Animals were assessed for sensorimotor deficits in a quiet and lowly lit (under red light) room during the last hour of the dark period (between 9 a.m. to 10 a.m.). After the assessment, the scores of each individual test parameter (0–3) were summed up. The lower score indicates higher sensorimotor deficit and vice-versa. The tests were performed by the investigators blinded to the treatment groups.

#### 2.3.2. Histology

To assess cerebral damage, and its relationship with sensorimotor deficit following MCAO, triphenyl tetrazolium chloride (TTC) staining was performed, as previously described [30]. After 1 h of MCAO, animals were examined for sensorimotor deficits (as described above) prior to euthanizing them at 3, 6, 12, 24, and 48 h (N = 5/time point) after reperfusion by decapitation under deep CO_2_ anesthesia. The brains were isolated in cold conditions using ice-cold phosphate-buffered saline (1X; PBS; pH 7.4; Leinco, St. Louis, MO, USA) for 1–2 min. Subsequently, under freezing conditions, coronal brain sections of approximately 1 mm thickness were obtained antero-posteriorly (AP) between AP +1.5 to −3 [31] using the adult mouse brain slicer matrices (Zivic, Pittsburgh, PA, USA). The sections were collected in a 15 mL falcon tube containing 0.05% TTC (Fisher Scientific, Hanover Park, IL, USA) solution at room temperature and incubated at 37 °C for 30 min under dark conditions with intermittent shaking. After 30 min, the sections were washed with PBS and stored in 10% buffered formalin until photographed. For image processing, Fiji was used, which is an open-source distribution of ImageJ (version 1.53q, National Institute of Health, Bethesda, MD, USA). The actual infarct volume adjusted for edema of each brain was measured in a blinded manner, as previously described [32]. We first calculated the infarcted area (IA) by subtracting the healthy area of the ipsilateral hemisphere (IL), identified using semi-automated color thresholding tools, from the contralateral hemisphere (CL) area, and then infarct volume was calculated by multiplying IA with the section thickness. The infarct percentage was calculated as a % of the contralateral hemisphere.

#### 2.3.3. Statistics

Pearson’s correlation was performed to examine the linear relationship between sensorimotor deficit and infarction %.

### 2.4. Experiment 2

Experiment 2 was designed to investigate sleep–wakefulness in the mouse MCAO model of IS. Sleep-recording electrodes were used to monitor sleep–wakefulness (sleep–W) in the animals and spectral analysis, as previously described [33]. Three bare-ended wire electrodes were implanted above the nuchal muscle to record EMG. Two stainless steel screw electrodes (EEG) were implanted on the skull using the following coordinates: right frontal [(AP: 1 mm; ML: −2 mm) and right posterior (AP: −2 mm; ML: −2 mm)] cortices. All the coordinates were relative to Bregma.

Subsequently, the mice were left undisturbed for post-operative recovery and habituation, with the recording set up for at least 5 days. The electrographic data were acquired and analyzed to determine the sleep onset latency, and power in different frequencies including delta (1–4 Hz) and/or theta (5–9 Hz), as previously described [33,34]. The stability of electrographic recording an d sleep–W cycle was verified by continuous, 48 h sleep–W recording and compared with the established sleep–W cycle in our laboratory. Subsequently, the animals were unhooked from the cables and subjected to MCAO, or sham (MCAO surgery without intraluminal insertion of filament) surgeries (N = 6/group) as described above. After recovery from surgical stress for at least 12 h, animals were again tethered to the sleep recording set-up, and their sleep–wakefulness was recorded and separately analyzed for light and dark periods starting from light onset on day 2 post-stroke.

#### Statistics

The student’s *t*-test was performed to determine the effects of IS on parameters of quality (sleep latency and delta power) and quantity (duration) of sleep–wakefulness.

### 2.5. Experiment 3

Experiment 3 was performed to understand whether sleep homeostasis is affected after IS. The animals were implanted with sleep-recording electrodes and subjected to MCAO or sham (N = 6/group) surgeries, as described above. On day 2 post-MCAO, using gentle handling techniques, animals were sleep deprived during the last 6 hours of the light period and left undisturbed, and allowed to recover sleep for the first three hours of the dark period as described previously [35,36,37]. Sleep–wakefulness was continuously monitored during sleep deprivation and recovery sleep.

#### Statistics

Two-way repeated measures ANOVA (RM ANOVA) (within subject = time (2 levels; first and last 2 h or 3 levels; first, middle, and last 2 h of sleep deprivation); between subject = treatment (2 levels; sham and MCAO)) were used to examine the effect of IS on markers of the building-up of sleep pressure (NREM sleep bouts and wake theta power) during sleep deprivation. An unpaired *t*-test was used to examine the effect of IS on latency to recovery sleep, amount of time spent in NREM sleep, and NREM delta power during the recovery sleep.

## 3. Results

### 3.1. Validation of MCAO Model

Garcia test followed by TTC staining was performed after 1 h of MCAO, and 3, 6, 12, 24, or 48 h of reperfusion (described above). The correlation analysis followed by *t*-test suggests that there is a significant correlation (r = −0.88; *p* = 0.0001; N = 25; Pearson correlation) between the Garcia test scores and infarction % (Figure 1). The mean ± SEM of the score is14.0 ± 0.3 (N = 5) at 3 h, 12.8 ± 0.4 (N = 5) at 6 h, 9.0 ± 0.4 (N = 5) at 12 h, 6.8 ± 0.4 (N = 5) at 24 h, and 5.0 ± 0.3 (N = 5) at 48 h. The mean ± SEM of the infarction % is 23.0 ± 3.7 (N = 5) at 3 h, 42.8 ± 3.5 (N = 5) at 6 h, 54.8 ± 1.4 (N = 5) at 12 h, 66.4 ± 3.3 (N = 5) at 24 h, and 73.2 ± 2.8 (N = 5) at 48 h.

### 3.2. IS Causes Insomnia-like Symptoms in Middle-Aged Mice

#### 3.2.1. During 24 h

Analysis of 24 h of sleep–wakefulness on day 2 post-MCAO reveals that, compared to the sham group (N = 6), IS mice (N = 6) display insomnia-like symptoms, as evident by significant increase in total amount of time spent in wakefulness (t = 6.54, df = 10, *p* < 0.001), and a reduction in the total amount of time spent in NREM (t = 4.49, df = 10, *p* < 0.01) and REM (t = 8.56, df = 10, *p* < 0.001) sleep (Figure 2A).

#### 3.2.2. During Light Period

Analysis of sleep–wakefulness during the light period of day 2 post-MCAO reveals that, compared to the sham group (N = 6), mice subjected to MCAO (N = 6) display a significant increase (t = 14.46, df = 10, *p* < 0.001, Figure 2B) in the latency to NREM sleep and a reduction (t = 4.46, df = 10, *p* < 0.01, Figure 2C) in the NREM delta power, suggesting that the quality of sleep is affected following IS. In addition, IS mice display a significant increase in wakefulness (t = 11.8, df = 10, *p* < 0.001) along with a reduction in NREM (t = 9.06, df = 10, *p* < 0.001) and REM (t = 9.36, df = 10, *p* < 0.001) sleep during the light period (Figure 2D), suggesting that the quantity of sleep is impaired following IS.

#### 3.2.3. During Dark Period

Analysis of sleep–wakefulness during the dark period of day 2 post-MCAO reveals that, compared to the sham group (N = 6), there is a significant reduction in the wakefulness (t = 2.97, df = 10, *p* < 0.05), with a concomitant increase in the NREM (t = 4.21, df = 10, *p* < 0.01) and REM (t = 3.82, df = 10, *p* < 0.01) sleep (Figure 2E) in stroke mice.

### 3.3. IS Disrupts Sleep Homeostasis

#### 3.3.1. Sleep Deprivation

During 6 hours of sleep deprivation, both the sham and MCAO (N = 6/group) groups spent the majority of time (>93%) in wakefulness (data not shown). Two-way RM ANOVA shows that, during sleep deprivation, there is a significant main effect of treatment (F (1,20) = 9.3; *p* = 0.012), time (F (1,20) = 76.11; *p* = 0.0001), and interaction (F (1,20) = 7.79; *p* = 0.003) on NREM sleep bouts. Further analysis using Bonferroni’s post-hoc test shows that, compared to the sham controls, stroke mice show a significantly (*p* < 0.05) lower number of NREM sleep bouts during the last two hours of sleep deprivation (Figure 3A), suggesting reduced sleepiness in stroke mice.

Two-way RM ANOVA shows that, during sleep deprivation, there is a significant main effect of treatment (F (1,10) = 10.52; *p* = 0.008), time (F (1,10) = 23.59; *p* = 0.0007), and interaction (F (1,8) = 7.94; *p* = 0.018) on theta power. Fine grain analysis using Bonferroni’s post-hoc test shows that mice in the sham group display a significant (*p* < 0.01) increase in theta power during the last two hours of sleep deprivation as compared to the first two hours; however, no increase in wake theta power is observed during the last two hours of sleep deprivation in the stroke group (Figure 3B). These results suggest that the mice in the stroke group do not show a building up of sleep pressure.

#### 3.3.2. Recovery Sleep

Compared to the sham controls, stroke mice take a significantly longer time to fall asleep (t = 3.49; df = 10; *p* = 0.0058; unpaired *t*-test; Figure 3C). In addition, stroke mice display a significant reduction in time spent in NREM sleep (t = 4.78; df = 10; *p* = 0.0007, unpaired *t*-test), along with a significant increase in the time spent in wakefulness (t = 4.2; df = 10; *p* = 0.02, unpaired *t*-test) during recovery sleep in the first three hours of the dark period. REM sleep remains unaffected (t = 0.76; df = 10; *p* = 0.46; unpaired *t*-test; Figure 3D). Furthermore, stroke mice display a reduction in the NREM delta power (t = 2.39; df = 10; *p* = 0.03; unpaired *t*-test) as compared to sham controls (Figure 3E) during three hours of recovery sleep.

## 4. Discussion

This is the first study that demonstrates that IS mice mimic sleep disturbances in stroke patients. Furthermore, IS-induced sleep disturbances are caused by impaired sleep homeostasis in this mouse MCAO model of IS.

In the present study, we used middle-aged C57BL/6J mice, and performed MCAO to mimic IS in humans [38,39,40]. The MCAO was performed in the middle of the light period to mimic circadian effects on clinical strokes while achieving circadian homogeneity [41,42,43]. Depending on the occlusion time, varying levels of lesion severity can be obtained in this model, however, 1 h occlusion in the MCAO model produces reproducible infarction in the MCA territories, including the frontoparietal cortex and lateral caudoputamen, and mimics human stroke symptoms [44,45]. We specifically targeted the right MCA in this study since sleep disturbances are commonly observed in patients with right hemispheric infarcts [46,47,48,49].

According to the American Academy of Sleep Medicine (AASM), insomnia is defined as difficulty in the initiation, duration, and/or quality of sleep during the night, resulting in daytime impairment [50]. In this study, we first examined the sleep–wakefulness across 24 h, and found that the IS mice spend significantly more time in wakefulness across the light–dark cycle (24 h). To further understand whether IS-induced sleep disruptions occur during the sleep (light) period, we analyzed the data separately for the light and dark periods. We found that during the normal sleep (light) period, mice have difficulty initiating sleep, evident by increased latency to sleep, reduction in the duration of sleep (NREM and REM), evident by a reduced amount of time spent in NREM and REM sleep, and reduction in the quality of sleep, evident by reduced NREM delta power.

In addition, similar to human stroke patients, IS mice display excessive daytime sleepiness-like symptoms during the dark (active) period, evident by reduced time spent in wakefulness along with an increased time spent in NREM sleep, reduced NREM latency, and a reduced amount of time spent in REM sleep [51,52,53,54,55].

Site-specific lesion studies suggest that lesions in the globus pallidus region produce a robust increase in wakefulness with a concomitant reduction in NREM sleep, suggesting insomnia-like symptoms. In addition, striatal (caudoputamen) lesions cause a reduction in wakefulness, suggesting daytime sleepiness [56]. Studies using different durations of MCAO in rat and mouse models of IS show that IS-induced infarction in cortex and striatum causes a reduction in NREM and/or REM sleep, although symptoms such as insomnia and excessive daytime sleepiness, similar to human IS patients, are not reported [57,58].

Previous studies observed that sleep homeostasis is compromised in stroke patients [49,59]. In our study, we observed that the overall wakefulness increases in the IS mice across a 24 h period, suggesting an impaired sleep homeostasis, but not circadian regulation. Impaired circadian regulation of sleep–wakefulness does not affect the total amount of time spent in wakefulness across 24 h. Instead, less wakefulness is observed during the dark (active) period and more wakefulness during the light (normal sleep) period. Thus, in our next set of experiments, we evaluated the effects of strokes on electrophysiological markers of sleep homeostasis.

Sleep deprivation followed by recovery sleep is an essential paradigm to study sleep homeostasis. During sleep deprivation, an increase in wake theta power and NREM sleep bouts are the indicators of sleep pressure build up (sleepiness). Dissipation of sleep pressure is evident by reduced sleep latency, and increased quality (delta power) and quantity of NREM sleep [60,61,62,63,64]. Our results suggest that mice with IS display impaired sleep homeostasis.

In conclusion, we used 1 h of MCAO to induce IS in middle-aged C57BL/6J mice and examined homeostatic regulation of sleep–wakefulness. Based on our results, we suggest that sleep disturbances observed in our mouse model of IS mimic sleep disturbances observed in human stroke patients. Furthermore, impaired sleep homeostasis may be causal to sleep disturbances observed in IS patients. Limitations of this study include (a) sex differences in this mouse model of IS and (b) circadian regulation of sleep–wakefulness in our mouse model of IS was not examined. However, future studies are planned to examine sex differences and circadian regulation in our IS mouse model.

## Figures and Tables

**Figure 1 cells-11-02818-f001:**
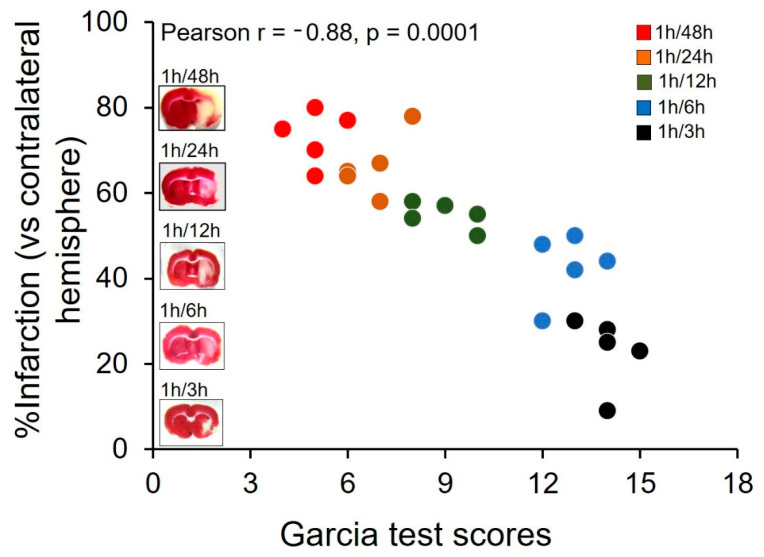
A correlation exists between sensorimotor deficit and %infarction in middle-aged mice subjected to 1 h of middle cerebral artery occlusion (MCAO) followed by reperfusion of 3, 6, 12, 24, and 48 h. The graph shows data points of each mouse (total N = 25) subjected to 1 h of MCAO to induce ischemic stroke (IS), denoting Garcia score (depicting sensorimotor deficit) and %infarction (depicting cerebral damage) examined at 3 (N = 5, denoted by black circles), 6 (N = 5, denoted by blue circles), 12 (N = 5, denoted by green circles), 24 (N = 5, denoted by orange circles), and 48 h (N = 5, denoted by red circles) post-stroke. Corresponding to each time point, a representative TTC-stained coronal brain section is shown to visualize the lesion. A significant correlation (*p* < 0.001) is found between Garcia score and %infarction assessed at different time points after MCAO.

**Figure 2 cells-11-02818-f002:**
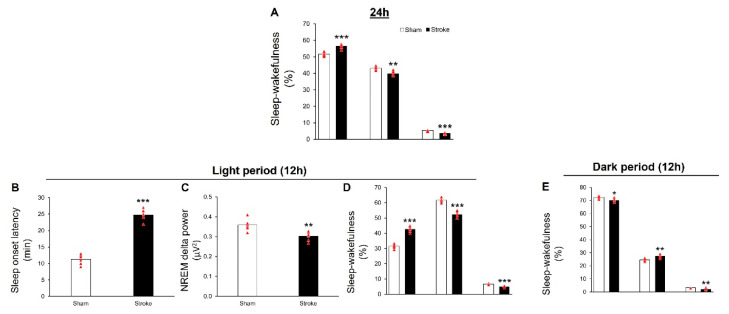
Ischemic stroke (IS) causes severe sleep disturbances in middle-aged mice, mimicking sleep in human IS patients. (**A**) describes the effect of IS, induced by 1 h of MCAO, on sleep–wakefulness during the 24 h period examined on day 2 post-MCAO. Compared to the sham group (N = 6), IS mice display a significant increase in wakefulness (*p* < 0.001), along with a reduction in NREM (*p* < 0.01) and REM (*p* < 0.001) sleep (N = 6), suggesting a disrupted compensatory sleep-loss recovery mechanism or impaired sleep homeostasis. (**B**–**D**) describes the effect of IS on sleep–wakefulness during the sleep (light) period. Compared to the sham group, IS mice (stroke group) display a significant reduction in the quality of sleep, as evident by increased sleep latency (*p* < 0.001 **B**) and a reduction in the NREM delta power (*p* < 0.001; **C**) and the quantity of sleep, as evident by increased time spent in wakefulness (*p* < 0.001) along with a reduction in NREM (*p* < 0.001) and REM (*p* < 0.001) sleep (**D**) sleep, suggesting insomnia-like symptoms. (**E**) Describes the effect of IS on sleep–wakefulness during the active (dark) period. Compared to the sham group, IS mice display a significant reduction in wakefulness (*p* < 0.05), along with an increase in NREM (*p* < 0.01) sleep, suggesting excessive daytime sleepiness symptoms. However, similar to the sleep period, the amount of time spent in REM sleep is found to be significantly (*p* < 0.01) reduced in the stroke group as compared to sham mice. All data are presented as mean ± SEM. * = *p* < 0.05; ** = *p* < 0.01 and *** = *p* < 0.001 vs. sham.

**Figure 3 cells-11-02818-f003:**
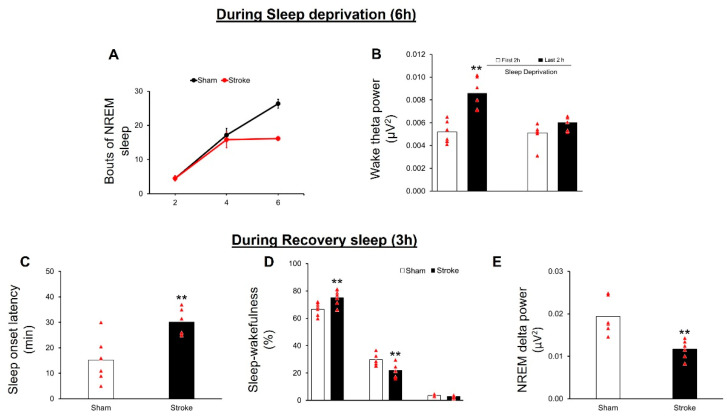
IS disrupts sleep homeostasis. (**A**): compared to sham controls, stroke mice display a significant (*p* < 0.05) reduction in the frequency of NREM bouts during the last 2 h of sleep deprivation. The NREM sleep bouts are comparable between the groups during first four hours of sleep deprivation. (**B**): stroke mice do not show any significant increase in the theta power during the last two hours of sleep deprivation as compared to first two hours. In contrast, sham mice display a building up of sleep pressure, as evident by a significant (*p* < 0.01) increase in wake theta power during the last two hours as compared to the first two hours. During recovery sleep, mice in the stroke group display a significant (*p* < 0.01) increase in latency to recovery sleep (**C**), a significant (*p* < 0.01) reduction in NREM sleep, and a significant (*p* < 0.01) increase in wakefulness during recovery sleep. REM sleep remains unaffected (**D**) and there is a significant (*p* < 0.01) reduction in the NREM delta power (**E**) as compared to the controls. All data are presented as mean ± SEM. ** = *p* < 0.01 vs. sham.

## Data Availability

Not applicable.

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
