# Peer review of "Ischemic Stroke Disrupts Sleep Homeostasis in Middle-Aged Mice"

_cells, 2022, doi:10.3390/cells11182818_

Round 1

Reviewer 1 Report

The manuscript “Ischemic stroke disrupts sleep homeostasis in the middle-aged 2

Mice” discusses the effect of IS on sleep homeostasis. The authors have investigated whether sleep homeostasis is disrupted upon exposure to IS usingC57BL/6J mice and MCA occlusion. This reviewer is of the opinion that the work done is novel and has scientific importance. The experiments are well laid out and explained thoroughly. One minor change that would drastically improve the manuscript is to expand the introduction to engage the general reader.

Author Response

We thank the editors and reviewers for reviewing our manuscript and providing us with constructive critiques. Our response to editor/reviewers’ suggestions/critiques is described below. Reviewers’ comments are italicized followed by our response in bold.

Reviewer #1:

  1. One minor change that would drastically improve the manuscript is to expand the introduction to engage the general reader.

Our Response: We have revised the Introduction as suggested.

Reviewer 2 Report

The manuscript titled "Ischemic stroke disrupts sleep homeostasis in the middle-aged mice" investigated how ischemic stroke induced sleep disturbances using mid-aged mice to mimic mid-aged humans. I suggest to approve the publication after some minor changes.

1) Firstly, in every figure, I highly suggest to change the title of y axis to what it was instead of putting it on top of the graph and using "mean ± SEM" on y, which is very unclear presentation in my opinion. It could be stated at the end of figure caption to clarify that "all data were presented as mean ± SEM";

2). Instead of using box to represent results, I suggest to use individual data point in plotting graphs, so that it was easily tell that each data point came from one mouse in each group. At the same time, it avoids overlapping with error bar like in figure 3B.

Also, with individual data presented, it is easier to tell if the data in each group are normal.

3). There is a single data point in Figure 3B, under the box of Stoke group "first 2h". Please double check if it is an error.

4). Since most of the data are compared using Student's t-test, is the normality test conducted in each comparison to make sure that Student's t-test is the most powerful one for comparing.

5). Are the ischemic stroke surgery work conducted around the same time during their circadian clock? Since there is research shows that time of stroke might affect neuronal survival:

Beker, Mustafa Caglar, et al. "Time-of-day dependent neuronal injury after ischemic stroke: implication of circadian clock transcriptional factor Bmal1 and survival kinase AKT." Molecular neurobiology 55.3 (2018): 2565-2576.

Overall, the paper provided interesting information on how sleep disturbance was induced by IS, while so far most research stated how sleep disturbance affected IS recovery. After the questions were answered, I believe that it is good to publish.

Thanks very much!

Author Response

We thank the editors and reviewers for reviewing our manuscript and providing us with constructive critiques. Our response to editor/reviewers’ suggestions/critique is described below. Reviewers’ comments are italicized followed by our response in bold.

Reviewer #2:

  1. Firstly, in every figure, I highly suggest to change the title of y axis to what it was instead of putting it on top of the graph and using "mean ± SEM" on y, which is very unclear presentation in my opinion. It could be stated at the end of figure caption to clarify that "all data were presented as mean ± SEM".

Our Response: We have revised all the graphs as suggested.

  1. Instead of using box to represent results, I suggest to use individual data point in plotting graphs, so that it was easily tell that each data point came from one mouse in each group. At the same time, it avoids overlapping with error bar like in figure 3B. Also, with individual data presented, it is easier to tell if the data in each group are normal.

Our Response: The data is now shown in bar graphs with individual data points.

  1. There is a single data point in Figure 3B, under the box of Stoke group "first 2h". Please double check if it is an error.

Our Response: Please see our response to the comment#2.

  1. Since most of the data are compared using Student's t-test, is the normality test conducted in each comparison to make sure that Student's t-test is the most powerful one for comparing.

Our Response: Yes, normality test was performed prior to statistical comparison. It is now clearly described in the text (see Page 2 Line 82).

  1. Are the ischemic stroke surgery work conducted around the same time during their circadian clock? Since there is research shows that time of stroke might affect neuronal survival.

Our Response: Yes. It is now clearly described in the text (see Page 2 Line 94).